# Molecular Characterization and Positive Impact of Brassinosteroids and Chitosan on *Solidago canadensis* cv. Tara Characteristics

**Iman M. El-Sayed [1], Rasha G. Salim [2], Eman F. El-Haggar [3], Rasha A. El-Ziat [4,\*]** and **Dina M. Soliman [1]**

1   Department of Ornamental Plants and Woody Trees, Agricultural and Biological Research Division, National Research Centre (NRC), Giza 12622, Egypt; imanelsayed6065@yahoo.com (I.M.E.-S.); dina123soliman@hotmail.com (D.M.S.)
2   Department of Microbial Genetic, Genetic Engineering and Biotechnology Research Division, National Research Centre, Dokki, Giza 12622, Egypt; rasha_gomma@yahoo.com
3   Nutrition and Food Science Department, National Research Centre, 33 El Bohouth St., Dokki, Giza 12622 Egypt; eman_nrc@hotmail.com
4   Ornamental Horticulture Department, Faculty of Agriculture, Cairo University, Giza 12613, Egypt
\*   Correspondence: rasha.mohamed@agr.cu.edu.eg

**Abstract:** Although goldenrod (*Solidago canadensis*) is considered an invasive plant in many countries, it is a global cut-flower species. In addition, demand for goldenrod has increased significantly in recent years. Thus, the present study aimed to evaluate the response of *Solidago canadensis* cv. Tara to brassinosteroids (BRs) at levels of $0.10^{-3}$, $10^{-6}$, and $10^{-8}$ M, and chitosan at 0, 100, 150, and 200 mg/L as a foliar application to increase the quality and quantity of production, and its polyphenolic compounds. Moreover, antibacterial activity and genetic polymorphism for both untreated and the optimally treated goldenrod were investigated. The results showed that the highest mean of growth characteristics was found when plants were treated with BRs at $10^{-8}$ M, whereas the longer vase life was obtained using 200 mg/L chitosan. Furthermore, higher pigment values, N, P, K, and total phenolic content, antioxidant capacity, chlorogenic acid, and rutin content were detected on plants treated with 200 mg/L chitosan. In addition, foliar application with 200 mg/L chitosan caused higher antibacterial activity among the control and BRs. The optimal treatment of BR at $10^{-8}$ M (89%) showed a low genetic similarity, based on sequence-related amplified polymorphism (SRAP) analysis, comparable with the control and 200 mg/L chitosan. BR at $10^{-8}$ M and 200 mg/L chitosan showed a significant enhancement of growth parameters. As a result, it can be concluded that goldenrod, as a herb extract, shows significant promise as a natural preservative in pharmaceutical, food, and cosmetic products.

**Keywords:** goldenrod; growth; cut flower; antioxidants; antibacterial; polymorphism; SRAP

## 1. Introduction

*Solidago canadensis* cv. Tara, a member of the Asteraceae family and commonly known as Canadian goldenrod, grows readily as a wildflower in North America, Europe, and Asia (Japan, China, Taiwan, and Russia) [1]. It is an excellent cut flower for arrangements and bouquets, with high longevity post-harvest, and is a landscaping flowering plant [2]. In addition, goldenrod has been used in traditional medicine for centuries as an antiphlogistic therapy, in urological, analgesic, febrifuge and gastrointestinal tract treatments, in burn treatments, and in ulcer treatments and liver aids [3]. It possesses several biological activities, including antimicrobial, antioxidant, and antimutagenic

activities [4,5]. In previous research, the flavonoids, phenolic acids, saponins, terpenes and sterols of *Solidago* have been detected [6]. Goldenrod is a photoperiodic plant, and day length processing is vital to obtain full control of the harvest's timing. During the morphological growing period, night temperatures of 14 °C and at day temperatures of 18 °C are optimal [1].

Plant growth and flowering are dependent on the balance of plant growth regulators (PGRs), and plants rapidly respond to alterations of the hormone equilibrium [7]. Brassinosteroids (BRs) are a new group of naturally occurring phytohormones that stimulate growth. Furthermore, cauliflower is considered a natural source of brassinosteroids [8]. BRs are involved in several plant processes, such as cell division and expansion, xylem differentiation, pollen elongation, the forming of the pollen tube, leaf bending, the germination of seeds, and reproductive development. The exogenous application of BRs has resulted in a wide variety of disease resistance [9].

Recently, the search for biological approaches to reduce the use of chemical products in agriculture has led to research into the use of biopolymer-based materials [10]. Chitosan is one of these biomaterials. Chitosan is an organic nutrient that includes macro- and micro-nutrients and plant hormones that increase plant antibodies. Furthermore, it results from the processing of the skin waste of crabs, shrimp shells, molds and others [11]. Chitosan is available in large quantities from the deacetylation of chitin, and has multiple advantages, including being environmentally friendly and low-cost. In addition, it easily combines with other compounds to achieve better performance. Numerous studies indicate that chitosan shows potential in the development of sustainable agriculture practices, and in food production and conservation [10].

Molecular markers are considered important tools for the detection of genetic diversity among plant species, and numerous molecular marker techniques are used to observe genetic diversity, for example, random amplified polymorphic DNA (RAPD), simple sequence repeat (SSR) [12], inter simple sequence repeat (ISSR) [13], and amplified fragment length polymorphism (AFLP) [14]. Sequence-related amplified polymorphism (SRAP) is a powerful and simple tool with higher reproducibility and a higher throughput scale than RAPD, and it is easier to perform than AFLP. Multiple co-dominant loci can be identified by the SRAP technique and can easily trap open reading frames (ORFs) [15]. Two-primer amplifications are based on SRAP markers and the length of each primer is 17 or 18 nucleotides. At the 5′ end, the primers have a core sequence (13 or 14 bases), and the first 10 or 11 bases are non-specific; CCGG follows these non-specific nucleotides in the forward primer and AATT in the reverse primer. At the 3′ end, the three nucleotides are selective.

The main objective of the present research was to investigate the importance of the application of brassinosteroids and chitosan to improve the quantitative and qualitative characteristics of *Solidago canadensis*, including chemical composition, phenolic compound content, antioxidant activity, and antibacterial activity. In addition, SRAP markers were used to establish the existence of the molecular genetic variation level, as a result of the ideal treatment of the morphological characteristics of *Solidago canadensis* cv. Tara.

## 2. Material and Methods

The experimental study was performed in two successive seasons, from January to June 2018 and replicated during the same months of 2019, at the Department of Ornamental Plants and Woody Trees and Microbial Genetics, in an open greenhouse at the National Research Center (NRC), Giza, Egypt. The study was performed to examine morphological and flowering characteristics, including chemical composition, antioxidant activity, and antimicrobial activity, and detect the genetic variation in the growth of *Solidago canadensis* cv. Tara.

### 2.1. Planting Material

*Solidago canadensis* cv. Tara seedlings were obtained from the commercial farm "Floramix Farm", Giza, Egypt. Seedlings with length 7 cm and nine to ten leaves per seedling were planted in clay pots

(25 cm); the soil comprised clay and sand at 1:1 v/v, and two seedlings were planted in each pot. Soil analysis was performed at the soil testing laboratory, National Research Centre (NRC) (Table 1).

**Table 1.** Soil analysis.

| pH | EC dS/m * | Soluble Cations Meq/L | | | | Soluble Anions (Meq/L) * | | | Soil Texture |
|----|-----------|------|------|------|------|----------------------|------|------|--------------|
| | | Ca | Mg | Na | K | CO$_3$ HCO$_3$ | Cl | SO$_4$ | |
| 7.88 | 0.492 | 1 | 0.6 | 3.1 | 0.2 | - | 3 | 1.9 | Sandy loam |

* Electrical conductivity (EC); deciSiemens per meter (dS/m); Milliequivalents per liter (Meq/L).

## 2.2. System of Lighting

Goldenrod plants were grown under normal temperatures (22–23 °C) and regulated day lengths of 16–18 lighting h/day using Tungsten lamps to prolong the day length from 9 PM to 3 AM (at a rate of 15 watts m$^{-2}$) through cyclic lighting of 15 min on and 15 min off. The lamps were mounted 2 m from the surface of the soil. The plants of the goldenrod remain in rosette form when the minimum temperature and day length are below 15 °C and 12 h, respectively; the influence of day length over the formation of the rosette is greater than the influence of cold temperature. Through the use of lighting and heating, a program of year-round production can be implemented [16]. Lighting was stopped when the stalks reached the targeted stem length of 30–40 cm. *Solidago* plants then grow generatively and form flowers.

## 2.3. Experiments Treatments

Brassinosteroid (BR) concentrations ($10^{-3}$, $10^{-6}$, and $10^{-8}$ M) and chitosan (CHT) (100, 150, and 200 mg/L) were used as a foliar spray application, applied three times in the morning until the running off point was reached. The first spray was applied 21 days after planting, and then two applications were made at two-week intervals.

## 2.4. Morphological Data

In this research, plant height (cm), number of leaves, total leaf area (cm$^2$)/plant (leaf area was measured using the following formula [17]: Leaf area (cm$^2$) = leaf dry weight (g) × disk area (cm$^2$)/disk dry weight (g)), stem diameter (cm), fresh weight of herb (FW) (g), and dry weight of herb (DW) (g) were determined.

## 2.5. Flowering Data

Number of inflorescences, length of inflorescence stalks (cm), days to flower (days), vase life (day), fresh weight of flowers (g), and dry weight of flowers were determined.

## 2.6. Pigment Content

The leaf samples (mg/g) fresh weight (F.W.) were ground to powder using a mortar and extracted with 85% methanol. Then, the extracts were centrifuged for 10 min at 8000 rpm. Photosynthetic pigments were assayed for the obtained supernatant according to a previous protocol [18] using a UV–Visible spectrophotometer (UV-1280, Shimadzu, Japan). The equations and specific absorption in the wavelength were 660, 640, and 440 nm to determine chlorophylls a and b, and carotenoids, respectively. Average values for the two seasons are presented in the results section.

## 2.7. Nitrogen Determination (%)

In this study nitrogen content was obtained from dried herb and the presented values are the means of two seasons. Nitrogen content was measured by the modified Kjeldahl method as described by Cottenie [19].

### 2.8. Phosphorus Determination (%)

Phosphorus content was assessed in dried herb using the ammonium molybdate method according to Snell [20], and the values are the averages of both seasons.

### 2.9. Potassium Determination (%)

Potassium content was determined in digested solution using a flame photometer according to Chapman [21]. Results are represented as g/100 g dry weight (D.W.) of the herb, and the values are the means of two seasons.

### 2.10. Bioactive Compounds

#### 2.10.1. Preparation of *Solidago canadensis* Plant Extracts

Plant leaves were washed with distilled water, cut into small sections, and dried in an oven at 40 °C for 48 h, then ground by a mill, sieved to the obtained particle size of ~20 meshes and maintained in air-tight plastic bags at room temperature. Finely powdered dried plant leaves were extracted with methanol (10%). The extracts were stored at 25 °C for 24 h then filtered through two layers of cheesecloth and centrifuged at 5000 rpm for 15 min at 4 °C. The supernatants of extracts were individually concentrated by a rotary evaporator to dry under reduced pressure.

#### 2.10.2. Determination of Total Phenolic Content

The total phenolic content (TPC) of the *Solidago canadensis* plant powder was determined colorimetrically using Folin–Ciocalteau reagent (Sigma-Aldrich, Steinheim, Germany) as described by Lafka [22]; values are the average of both seasons. TPC was calculated using the gallic acid (Sigma Aldrich, Germany) calibration curve and are expressed as mg gallic acid equivalent (GAE/g).

#### 2.10.3. Determination of Antioxidant Activity

The antioxidant activities of the *Solidago canadensis* plant methanolic extracts were evaluated using the 2,2-diphenyl-1-picryl-hydrazyl (DPPH) radical scavenging method according to Matthus [23]; values are the mean of two seasons. Antioxidant activity is expressed as the percentage inhibition of DPPH, calculated according to the following formula:

$$(\%)\ \text{Inhibition} = ((A^{\text{Control}} - A_0^{\text{Sample}})/A^{\text{Control}}) \times 100$$

where $A$ is the absorbance at 517 nm of the control sample and $A_0$ is the final absorbance of the test sample at 517 nm.

#### 2.10.4. Identification of Phenolic Compounds of Solidago Extracts Using HPLC

Phenolic components of goldenrod methanolic extracts were identified and quantified by HPLC using an Agilent 1260 series. The separation was carried out using a Kromasil C18 column (4.6 × 250 mm i.d., 5 µm); the column temperature was maintained at 35 °C. The mobile phase consisted of water (A) and 0.05% trifluoroacetic acid in acetonitrile (B) with a flow rate of 1 mL/min; it was programmed consecutively in a linear gradient as follows: 0 min (82% A); 0–5 min (80% A); 5–8 min (60% A); 8–12 min (60% A); 12–15 min (85% A) and 15–16 min (82% A). The multi-wavelength detector was monitored at 280 nm. The injection volume was 10 µL for each sample solution. The separation was carried out using a Kromasil C18 (Scantec Lab, Sävedalen, Sweden) column (4.6 mm × 250 mm i.d., 5 µm), and the column temperature was maintained at 35 °C.

### 2.11. Antibacterial Activity

*Solidago canadensis* extracts in the concentrations of 12.5, 25, and 50 mg/mL, using streptomycin antibiotic with a concentration of 30 μg/L as the positive control, were derived for the agar diffusion assay [24]. Cultures of *Bacillus cereus*, *Pseudomonas aeruginosa*, *P. fluorescens*, *Xanthomonas. campestris* and *Escherichia. coli* were grown exponentially in nutrient broth at 37 °C for 18 h. The culture suspension was mixed with the prepared medium and poured into plates. Wells with a diameter of 0.5 mm were made and the test compound was applied individually to each well. Then, the plates were incubated for 24 h at 37 °C. The inhibition diameter zone in mm was measured.

### 2.12. Sequence-Related Amplified Polymorphism (SRAP)

#### 2.12.1. SRAP-PCR Reactions

A set of twelve companion primers was used to detect the genetic polymorphism, as shown in Table 2. The amplification reaction was carried out in a 25 μL reaction volume containing 12.5 μL Master Mix (0), 1.5 μL forward primer, 1.5 μL reverse primer (10 pcmol), 2.5 μL template DNA (10 ng), and 7 μL dH2O. The amplification was carried out in a DNA thermocycler (MWG-BIOTECH Primus) programmed for a denaturation cycle for 3 min at 94 °C, and 5 cycles were run at 94 °C for 40 s, 35 °C for 50 s and 72 °C for 1 min, for denaturing, annealing, and extension, respectively. Thirty-five cycles were run for a denaturation step at 94 °C for 1 min, an annealing step at 50 °C for 1 min, and an elongation step at 72 °C for 1.5 min. The primer extension segment was extended to 7 min at 72 °C in the final cycle. The amplification products were analyzed by electrophoresis in a 1% agarose gel stained with ethidium bromide and photographed using a gel documentation system (BIO-RAD 2000).

**Table 2.** Sequence-related amplified polymorphism (SRAP) primers used in this study.

| Primer | Forward Primer | Reverse Primer |
| --- | --- | --- |
| SRAP-1 | ME1-5′-TGAGTCCAAACCGGATA-3′ | EM1-5′-GACTGCGTACGAATTAAT-3 |
| SRAP-2 | ME1-5′-TGAGTCCAAACCGGATA-3′ | EM2-5′-GACTGCGTACGAATTTGC-3′ |
| SRAP-3 | ME1-5′-TGAGTCCAAACCGGATA-3′ | EM3-5′-GACTGCGTACGAATTGAC-3′ |
| SRAP-4 | ME1-5′-TGAGTCCAAACCGGATA-3′ | EM4-5′-GACTGCGTACGAATTTGA-3′ |
| SRAP-5 | ME2-5′-TGAGTCCAAACCGGAGC-3′ | EM1-5′-GACTGCGTACGAATTAAT-3 |
| SRAP-6 | ME2-5′-TGAGTCCAAACCGGAGC-3′ | EM2-5′-GACTGCGTACGAATTTGC-3′ |
| SRAP-7 | ME2-5′-TGAGTCCAAACCGGAGC-3′ | EM3-5′-GACTGCGTACGAATTGAC-3′ |
| SRAP-8 | ME3-5′-TGAGTCCAAACCGGAAT-3′ | EM1-5′-GACTGCGTACGAATTAAT-3 |
| SRAP-9 | ME3-5′-TGAGTCCAAACCGGAAT-3′ | EM2-5′-GACTGCGTACGAATTTGC-3′ |
| SRAP-10 | ME3-5′-TGAGTCCAAACCGGAAT-3′ | EM3-5′-GACTGCGTACGAATTGAC-3′ |
| SRAP-11 | ME4-5′-TGAGTCCAAACCGGACC-3′ | EM1-5′-GACTGCGTACGAATTAAT-3 |
| SRAP-12 | ME4-5′-TGAGTCCAAACCGGACC-3′ | EM2-5′-GACTGCGTACGAATTTGC-3′ |

#### 2.12.2. Data Analysis

To determine the genetic relatedness of the samples under study, the banding patterns generated by the SRAP markers were analyzed and compared. The products with clear and distinct amplification were scored as 1 for the presence of bands and 0 for their absence. Bands of the same mobility were scored as identical. The genetic similarity coefficient (GS) between two genotypes was estimated according to the Dice coefficient [25].

### 2.13. Statistical Analysis

The data were analyzed using a randomized complete block design with 3 replicates per treatment. The treatment averages were compared for significance by new multiple range tests at a 0.05% level of probability [26] using COSTATV-63.

## 3. Results and Discussion

### 3.1. Morphological Characteristics

The data presented in Table 3 indicate the influence of applying brassinosteroids at the levels of 0, $10^{-3}$, $10^{-6}$, and $10^{-8}$ M, and chitosan at the levels of 0, 100, 150, and 200 mg/L, on the vegetative growth of goldenrod plants compared with untreated plants. The results show that most morphological parameters, i.e., plant height, the number of leaves, stem diameter, total leaf area, herb fresh weight, and herb dry weight, were significantly changed. The data presented in Table 3 show that brassinosteroids significantly increased most of the vegetative parameters gradually by decreasing the dose of brassinosteroids. In contrast, the plants sprayed with chitosan significantly increased most of the vegetative parameters gradually by increasing the dose of chitosan. BR at $10^{-8}$ M yielded the highest values of plant height (76.16 and 72.03 cm), number of leaves (58.00 and 64.00), stem diameter (0.60 and 0.58 cm), herb fresh weight (14.97 and 12.79 g), herb dry weight (8.11 and 6.81 g), and total leaf area (690.43 and 586.58 cm$^2$) in the first and second season, respectively, compared to control and the other treatments studied.

**Table 3.** Effects of different concentrations of brassinosteroids and chitosan on the morphological characteristics of *Solidago canadensis* cv. Tara plants during 2018 and 2019.

| Characters \ Treatments | First Season | | | | | |
|---|---|---|---|---|---|---|
| | Plant Height (cm)/Plant | No. of Leaves/Shoot | Stem Diameter (cm)/Plant | Total Leaf Area/Plant (cm²) | Herb F.W (g) | Herb D.W (g) |
| Control | 53.50 e | 46.00 c | 0.40 d | 271.66 f | 4.25 f | 1.88 d |
| BRs $10^{-3}$ M | 68.67 b,c | 52.00 b,c | 0.50 b,c | 444.20 d | 6.16 e | 2.96 c |
| BRs $10^{-6}$ M | 70.39 b | 55.00 a,b | 0.56 a,b | 505.76 b | 10.33 b,c | 4.86 a,b |
| BRs $10^{-8}$ M | 76.16 a | 58.00 a | 0.60 a | 690.43 a | 11.30 a | 5.56 a |
| CHT100 mg/L | 60.15 d | 48.00 c | 0.42 d | 339.15 e | 8.62 d | 4.02 b,c |
| CHT 150 mg/L | 64.50 c | 51.00 b,c | 0.45 c,d | 444.48 d | 9.78 c | 4.59 a,b |
| CHT 200 mg/L | 72.07 a,b | 55.00 a,b | 0.57 a | 523.35 c | 11.02 a,b | 4.92 a,b |
| Characters \ Treatments | Second Season | | | | | |
| | Plant Height (cm)/Plant | No. of Leaves/Shoot | Stem Diameter (cm)/Plant | Total Leaf Area/Plant (cm²) | Herb F.W (g) | Herb D.W (g) |
| Control | 48.01 e | 43.00 b | 0.39 c | 152.83 g | 3.37 e | 1.70 d |
| BRs $10^{-3}$ M | 65.05 b,c | 50.00 b | 0.50 b | 352.13 d | 5.18 d | 2.70 c,d |
| BRs $10^{-6}$ M | 66.48 a,b | 53.00 a,b | 0.54 a,b | 503.59 b | 9.37 a,b | 4.67 a,b |
| BRs $10^{-8}$ M | 72.03 a | 64.00 a | 0.58 a | 586.58 a | 10.89 a | 5.41 a |
| CHT100 mg/L | 59.15 c | 44.00 b | 0.40 c | 207.06 f | 6.88 c | 3.60 b,c |
| CHT 150 mg/L | 55.77 d | 46.00 b | 0.40 c | 274.29 e | 8.62 b | 4.35 a,b |
| CHT 200 mg/L | 69.03 a,b | 53.00 a,b | 0.54 a,b | 381.2 c | 10.01 a,b | 5.01 a |

Averages (means) in each column with the same letter (s) are not significantly different according to Duncan 1955 test with Bonferroni correction ($p \leq 0.05$). Lower case letters (a–g) are for comparison of individual treatment means.

These results are in agreement with Badawy [27], who reported that *Zinnia elegans* plants treated with BR increased growth. In other studies, Leubner-Metzger [28] found that BR application enhanced the germination and increased seedling elongation of non-photo dormant tobacco seeds, and Castorina [29] reported that BR application had a significant role in promoting plant growth. In addition, Ellen [30] indicated that plants treated with chitosan had a significant number of secondary branches and a greater leaf canopy width than those without chitosan treatment. Salachna [31] showed that the addition of chitosan at 20 and 30 mL/L resulted in the best results compared to other treatments. The morphology of plants treated with chitosan administration was shown to have a more significant growth response than those without chitosan (control). Similarly, Salachna [32] noted that using medium- and high-molecular weight chitosan resulted in a positive increase in the corm weight, as a result of foliar spray treatment with high-molecular weight chitosan on freesia plants. The action of brassinosteroids is vital for plant growth and improvement. They modulate gene expression and control a vast range of processes, including the elongation and division of the cell, vascular differentiation, plant growth and reproductive development [33].

The chitosan plant growth promoter raises the photosynthesis level in the development of photosynthesis, thus aiding in stem enlargement. The increase in the diameter of the stem is due

to the relatively healthy growth of the plant, which occurs due to the adequate availability of nutrients. The higher rate of photosynthesis is an indicator of positive growth. More carbohydrates are translocated into the phloem and can be used to enhance secondary growth, which leads to the expansion of stem cells and the diameter of the stem [34]. Furthermore, the use of chitosan stimulates the activity of nitrogen metabolism enzymes (glutamine synthetase, nitrate reductase, and protease) that promote plant development and growth in rice. Similar results also show that the application of chitosan in okra plants increases total dry mass [34,35].

### 3.2. Flowering Parameters

In the present study, all of the flower parameters increased gradually with a decrease in the rate of brassinosteroids from $10^{-3}$ M to $10^{-8}$ M., whereas the flower parameters increased gradually with the increasing chitosan from 150 to 200 mg/L. The highest significant values of the flower parameters (Table 4) number of inflorescences, length of inflorescence stalks and flowers' fresh and dry weight in the first and second seasons were detected in plants sprayed with brassinosteroids at $10^{-8}$ M in comparison to untreated and other treatments. The results of the present study also show that chitosan accelerates flowering time. The chitosan at 200 mg/L yielded the best acceleration in flowering time (after 107.18 and 104.44 days) in the first and second seasons, respectively.

**Table 4.** Effect of different concentrations of brassinosteroids and chitosan on flowering characteristics of *Solidago canadensis* cv. Tara plant during 2018 and 2019.

| Characters<br>Treatments | First Season | | | | | |
|---|---|---|---|---|---|---|
| | No of<br>Inflorescence/Plant | Length of Inflorescence<br>Stalks (cm) | Days to<br>Flower (Days) | F.W.<br>Flowers/Plant (g) | D.W.<br>Flowers/Plant (g) | Vase<br>Life/Day |
| Control | 19.00 [d] | 65.60 [g] | 115.72 [a] | 4.25 [f] | 1.88 [d] | 8.66 [e] |
| BRs $10^{-3}$ M | 20.00 [c,d] | 74.50 [f] | 115.52 [a] | 6.16 [e] | 2.96 [c] | 10.03 [d,e] |
| BRs $10^{-6}$ M | 28.00 [b] | 97.02 [c] | 110.25 [c] | 10.33 [b,c] | 4.86 [a,b] | 11.22 [d] |
| BRs $10^{-8}$ M | 35.00 [a] | 112.02 [a] | 109.24 [c,d] | 11.30 [a] | 5.56 [a] | 13.03 [c] |
| CHT100 mg/L | 23.00 [c] | 86.66 [e] | 113.12 [b] | 8.62 [d] | 4.02 [b,c] | 14.22 [c] |
| CHT 150 mg/L | 27.00 [b] | 93.06 [d] | 111.22 [b,c] | 9.78 [c] | 4.59 [a,b] | 16.88 [b] |
| CHT 200 mg/L | 29.00 [b] | 106.51 [b] | 107.18 [d] | 11.02 [a,b] | 4.92 [a,b] | 20.11 [a] |
| Characters<br>Treatments | Second Season | | | | | |
| | No of<br>Inflorescence/Plant | Length of Inflorescence<br>Stalks (cm) | Days to<br>Flower (Days) | F.W<br>Flowers/Plant (g) | D.W<br>Flowers/Plant (g) | Vase<br>Life/Day |
| Control | 16.00 [d] | 55.04 [g] | 112.34 [a] | 3.37 [e] | 1.70 [d] | 7.88 [d] |
| BRs $10^{-3}$ M | 18.00 [c,d] | 63.35 [f] | 110.25 [b] | 5.18 [d] | 2.70 [c,d] | 8.91 [d] |
| BRs $10^{-6}$ M | 26.00 [b] | 86.05 [c] | 106.39 [c,d] | 9.37 [a,b] | 4.67 [a,b] | 10.14 [c,d] |
| BRs $10^{-8}$ M | 32.00 [a] | 105.09 [a] | 105.29 [d] | 10.89 [a] | 5.41 [a] | 12.71 [b,c] |
| CHT100 mg/L | 20.00 [c] | 78.20 [d] | 112.25 [a] | 6.88 [c] | 3.60 [b,c] | 13.34 [b,c] |
| CHT 150 mg/L | 25.00 [b] | 75.94 [e] | 108.12 [c] | 8.62 [b] | 4.35 [a,b] | 15.54 [b] |
| CHT 200 mg/L | 27.00 [b] | 93.55 [b] | 104.44 [d] | 10.01 [a,b] | 5.01 [a] | 19.02 [a] |

Averages (means) in each column with the same letter (s) are not significantly different according to Duncan 1955 test with Bonferroni correction ($p \leq 0.05$). Lower case letters (a–g) are for comparison of individual treatment means.

In agreement with our results, Pipattanawong [36] stated that the application of BRs increases the number of flowers in strawberry at the foliage stage. However, Rao [37] stated that in autumn, the use of BRs increased the number of flowers in grapes but inhibited the number if the time of use was delayed until late winter. Similarly, Salachna [32] found that using medium- and high-molecular weight chitosan resulted in a positive increase in the corm weight as a result of foliar spray treatment with high-molecular weight chitosan. No effect was observed on the length of the main inflorescence shoot or the length of inflorescence without chitosan. In addition, chitosan accelerated the time of flowering and increased the number of flowers in passion fruit plants [38]. Furthermore, it improved plant and root growth in vines, accelerated flowering time, and increased the fruit weight, yield and number of flowers in grapes [39].

Brassinosteroids play the main role in regulating the processes that lead to senescence. Brassinosteroids encourage senescence in *Xanthium* and *Rumex* explants [40], and in wheat leaves [41].

Chitosan is an organic nutrient which, in solution, contains macro- and micro-nutrients. Chitosan also contains production and growth hormones, including cytokinin (Zeatin), auxin (IAA) and gibberellins, to increase plant antibodies, and enhance growth and increase the yield [11].

### 3.3. Vase Life

In the current study, chitosan had a significant effect on the vase life of *Solidago* cut flowers. The data in Table 4 show that chitosan at 200 mg/L yielded the longest vase life of *Solidago* cut flowers (20.11 and 19.02 days) in the first and second season, respectively. In contrast, brassinosteroid treatment yielded the shortest vase life compared to chitosan treatment and the control. The control treatment resulted in the shortest vase life (8.66 and 7.88 days) in the first and second season, respectively, compared to other treatments used.

These findings were confirmed by Solgi [42], who noted that using 50 mg/L of chitosan in the vase solution increased the vase life of carnation cut flowers in comparison to the control. Another study reported that chitosan is used as a coating for seeds, leaves, fruits and vegetables to stimulate plant growth, increase plant production, and protect plants against microorganisms [43]. Furthermore, chitosan has been demonstrated to be an antimicrobial agent against pathogenic fungi, bacteria, and mold because of its cationic groups [44]. Nonetheless, few studies have been conducted on the effects of brassinosteroids (BRs) and chitosan (CHT) on the shelf life of cut flowers.

Senescence is the process of endogenously regulated deterioration resulting from alterations that naturally cause the death of organs, tissues, and cells, or of the whole organism [45]. BRs also play an important role in regulating the processes leading to the senescence process, like other hormones [37]. Brassinolides stimulate senescence in *Xanthium* and *Rumex* explants [40] and in wheat leaves [41]. In addition, BRs also accelerate senescence in mung bean leaves and seedlings [46].

The antimicrobial mechanism of chitosans is due to an interaction between the microorganisms and the cationic surface, causing alterations in the cell membrane and programmed cell death. Furthermore, scientists have recommended that chitosan (CHT) can block the transcription of RNA from DNA or act as a chelating agent, such as a metal, by suppression of the nutrients vital to bacterial and fungi growth [47]. Chitosan application has been demonstrated to preserve the quality and prolong the longevity of numerous fruits, such as peach, mango, strawberry, papaya, grape, and sweet cherry [47]. Nevertheless, few studies have investigated the effects of chitosan (CHT) on the shelf life of cut flowers.

### 3.4. Chemical Composition

### 3.4.1. Pigment Content

The results of the analysis of variance of the treatments are shown in Table 5. The applications of chitosan and brassinosteroids each had significant effects on chlorophyll a and b and the total carotenoids. The highest values of chlorophyll a (1.37 mg/g F.W.), chlorophyll b (0.47 mg/g F.W.) and total carotenoids (0.70 mg/g F.W.) were obtained in plants treated with chitosan at 200 mg/L, followed by plants treated with brassinosteroids at $10^{-8}$ M. In contrast, the lowest values were obtained in the control plants. These findings are in a good agreement with Zhang [48], who detected that the use of brassinosteroids enhanced the assimilation of carbon and nitrogen by the stabilization of membrane structures, and also improved plant growth, productivity and photosynthesis. Sadak [49] showed that use of brassinosteroids significantly increased chlorophyll a and b, carotenoid, and total photosynthetic pigments in quinoa plants. In addition, Salachna [32] noted that the use of medium- and high-molecular weight chitosan resulted in better plants with higher relative chlorophyll contents (SPAD).

BRs acting as a growth-stimulating substance can also play a role in improving growth and development via the stimulation of certain metabolic activities. BRs can also improve the rate of light saturated net $CO_2$ assimilation and the carboxylation of rubisco, thereby increasing the ability of $CO_2$ assimilation in the Calvin cycle [50].

Correspondingly, the chitosan promoter raises the photosynthesis level in the development of photosynthesis, thus aiding in stem enlargement. The increase in stem diameter is due to the relatively healthy growth rate of the plant, which results from the adequate availability of nutrients. Thus, the higher rate of photosynthesis is an indicator of good growth [51].

**Table 5.** Effects of different concentrations of brassinosteroids and chitosan on the chemical composition of *Solidago canadensis* L cv. Tara; the values are the mean of two seasons.

| Treatments | Chemical Composition | | | | | | | |
|---|---|---|---|---|---|---|---|---|
| | Chlorophyll [a] mg/g F.W. | Chlorophyll [b] mg/g F.W. | Carotenoids mg/g F.W. | N % | P% | K% | Total Phenolic (mg GAE/g Tissue) | Antioxidant Activity |
| Control | 0.65 [c] | 0.23 [e] | 0.30 [e] | 2.00 [g] | 0.20 [c] | 1.96 [d] | 45.76 [e] | 14.40 [e] |
| BRs $10^{-3}$ M | 0.82 [b,c] | 0.32 [b] | 0.46 [c] | 2.67 [d] | 0.22 [b,c] | 2.54 [b] | 63.11 [c] | 19.32 [c] |
| BRs $10^{-6}$ M | 0.92 [b] | 0.34 [b] | 0.52 [b] | 2.88 [c] | 0.24 [b] | 2.61 [b] | 66.68 [b] | 27.30 [b] |
| BRs $10^{-8}$ M | 1.35 [a] | 0.45 [a] | 0.68 [a] | 3.29 [b] | 0.27 [a] | 2.77 [a] | 71.47 [a] | 29.82 [b] |
| CHT100 mg/L | 0.70 [c] | 0.30 [b] | 0.31 [e] | 2.09 [f] | 0.20 [c] | 2.01 [d] | 55.77 [d] | 16.11 [d,e] |
| CHT 150 mg/L | 0.74 [c] | 0.30 [b] | 0.41 [d] | 2.29 [e] | 0.21 [c] | 2.15 [c] | 61.01 [c] | 18.79 [c,d] |
| CHT 200 mg/L | 1.37 [a] | 0.47 [a] | 0.70 [a] | 3.44 [a] | 0.29 [a] | 2.88 [a] | 72.92 [a] | 38.72 [a] |

Averages (means) in each column with the same letter (s) are not significantly different according to Duncan 1955 test with Bonferroni correction ($p \leq 0.05$). Lower case letters (a–g) are for comparison of individual treatment means.

### 3.4.2. Content of Macronutrients

The highest levels of N, P, and K content were obtained using 200 mg/L chitosan followed by $10^{-8}$ M of BRs, as shown in Table 5. The lowest mean value was found in plants sprayed with 100 mg/L chitosan. These results are in line with Sadak [49], who reported that brassinosteroids increased the nutritional value of the yield of seeds of the quinoa plant. Moreover, El-Tanahy [52] revealed that the best effect on total protein, total carbohydrates, and N, P, and K content was obtained using the highest concentration of chitosan (5%). Thus, there is a positive correlation between increasing the concentration of chitosan and the response of chemical composition and all growth characteristics.

### 3.4.3. Total Phenolic Content

The total phenolic content in goldenrod plants is shown in Table 5. It can be observed that the TPC of all goldenrod treatments was significantly higher than that of the control. TPC increased gradually with declining levels of brassinosteroids, whereas it increased gradually as the chitosan concentration increased. The highest TPC was found with 200 mg/L chitosan and BRs at $10^{-8}$ M, and the lowest TPC was found in untreated plants. This observation is similar to that of [53], in which the foliar application of brassinosteroids enhanced radical scavenging activity in plants. Bautista-Banos [54] found that chitosan had positive antimicrobial properties, and chitosan also has the potential to induce protection-related enzymes and phenolic content (TPC) in plants [55]. In addition, it was noted that the phenolic compounds in chitosan-treated goldenrod were significantly higher than those of the control. Chitosan application of 200 mg/L was the most active in increasing total phenolic compounds among all of the different treatments. This result is not consistent with Liu [56], who stated that the accumulation of phenolic compounds was induced in fruit and tomato plants treated with chitosan. Similarly, Salimgandomi [57] reported that the antioxidant activity of mint (*Mentha piperita* L.) can be stimulated with chitosan treatment. Furthermore, antioxidant activity and all phenolic contents were increased by increasing the concentrations of chitosan. The use of chitosan as a foliar spray results in a significant increase in secondary metabolites, particularly compounds of phenols and flavonoids, which may destroy the combination of hydrogen peroxide and free radicals, and thus increase antioxidant activity [57].

### 3.4.4. Antioxidant Activity

It can be noted in Table 5 that antioxidant activity significantly increased with a foliar application with 200 mg/L chitosan, followed by BRs at $10^{-8}$ M. The highest antioxidant capacity was detected in goldenrod plants treated with 200 mg/L chitosan. This higher antioxidant capacity could be due to the higher total phenolic content of goldenrod treatments. Previous studies have established a positive correlation between antioxidant activity and total phenolic content [58]. The high total antioxidant capacity may therefore be due to the high total phenolic content. Raghu [53] reported that the contents of phenols and flavonoids were improved due to brassinosteroid supplementation. Additionally, the foliar application of brassinosteroids enhances radical scavenging activity compared to untreated control plants, which could be due to increases in the phenolic, flavonoid, and pigment contents [59]. Furthermore, Yu [50] reported that a foliar spray of brassinosteroids significantly increased antioxidant compounds (flavonoids and phenolic compounds) and antioxidant activity in quinoa plants. Ghasemnezhad [60] noted that TPC and antioxidant activity were significantly increased with chitosan at the level of 0.5%. Moreover, chitosan significantly prolonged polyphenol and antioxidant activity in post-harvest strawberries, and flesh browning was inhibited under cold storage conditions [61].

### 3.4.5. Identification and Quantification of Individual Polyphenolic Compounds of *Solidago canadensis* L cv. Tara Extracts

The identified phenolic compounds in the *Solidago canadensis* extracts were quantified using HPLC as presented in Table 6. Overall, the results show that the contents of the polyphenolic compounds of BRs at $10^{-8}$ M and CHT 200 mg/L of goldenrod plant samples were higher than those of the control plant extract. Caffeic acid, coumaric acid and taxifolin were not detected in the control compared to the prepared samples using BRs at $10^{-8}$ M and CHT 200 mg/L. In addition, catechin and ferulic acid were not detected in all plant samples. CHT 200 mg/L samples of the goldenrod plant were rich in phenolic compounds and rutin, as a flavonoid glycoside was the polyphenolic component detected at the highest level (364.10 µg/g), whereas ellagic acid (4.82 µg/g) was detected only in CHT samples. Chlorogenic acid (276.20 µg/g), kaempferol (211.36 µg/g), gallic acid (178.13 µg/g), and naringenin (133.42 µg/g) polyphenolic constituents were also detected in CHT samples compared to other samples of goldenrod plant treated with BRs. However, rutin was the main phenolic component detected in *Solidago canadensis* extracts [62]. In addition, chlorogenic acid as a phenolic acid (207.26 µg/g) was the polyphenolic component most detected in the sample treated with BRs at $10^{-8}$ M, followed by gallic acid (134.92 µg/g). Chlorogenic acid is a plant secondary metabolite that is widely distributed in coffee, tea, and many fruits, vegetables, and herbs. It has multiple biological properties, including antioxidant, antioxidant, antibacterial, and antiviral properties [63]. In addition, chlorogenic acid has been used as a nutraceutical agent for the prevention and treatment of metabolic syndrome and associated diseases [64].

### 3.5. Antibacterial Activities

The potential antibacterial activity of the *Solidago canadensis* methanol extract (as the control) and two different treatments (BRs $10^{-8}$ M and CHT 200 mg/L) with concentrations of 12.5, 25 and 50 mg/mL was determined using an agar well diffusion assay. The highest significant antibacterial activity against *P. fluorescens* and *Bacillus cereus* among all tested bacteria was found in goldenrod plant extracts. In contrast, no activity was found against *X. campestris* and *Pseudomonas aeruginosa*, and low antibacterial activity was found against *E. coli*; the results are presented in Figures 1 and 2. The significant concentrations in the control and treatments were 50 mg/mL. Numerous studies have shown that polyphenolic compounds have a significant antibacterial activity [65], and regarding the results obtained for the phytochemical characterization of *S. canadensis* extracts, it can be stated that the higher quantity of polyphenols from methanol extract can result in better antibacterial activity. Furthermore, chlorogenic acid has shown strong antimicrobial activity against many bacterial strains

(*Escherichia coli*, *S. aureus*, *P. aeruginosa.* and *Klebsiella pneumoniae*, [66], and *H. pylori.* [67]). Thus, it could be used as a preservative in pharmaceutical or cosmetic products, and as a food additive [63]. Another study reported that *S. canadensis* essential oil has strong antimicrobial activity, mainly due to its high content of active secondary metabolites, such as terpenoids, flavonoids, polysaccharides, and phenolic compounds [68]. Few studies on the biological effects of essential oils and vegetal extracts of *S. canadensis* have been conducted. Alves [65] concluded that various extracts of *S. canadensis* showed promising antibacterial activity compared with ciprofloxacin, as a commercial antibiotic against pathogenic bacteria, such as *Salmonella typhi*.

**Table 6.** Polyphenolic composition of *Solidago canadensis* cv. Tara extract by HPLC quantification analysis.

| Polyphenolic Compound | Polyphenolic Compounds Content of *Solidago canadensis* cv Tara Extracts (µg/g) | | |
|---|---|---|---|
| | Control * | BRs * $10^{-8}$ M | CHT *200 mg/L |
| Gallic acid | 5.50 | 134.92 | 178.13 |
| Chlorogenic acid | 24.58 | 207.26 | 276.20 |
| Catechin | ND * | ND * | ND * |
| Methyl gallate | 4.54 | 15.27 | 31.27 |
| Caffeic acid | ND * | 9.90 | 15.57 |
| Syringic acid | 4.21 | 34.84 | 50.25 |
| Pyro catechol | 8.36 | 32.26 | 73.46 |
| Rutin | 24.71 | 170.05 | 364.10 |
| Ellagic acid | ND * | ND * | 4.82 |
| Coumaric acid | ND * | 5.23 | 5.30 |
| Vanillin | 3.20 | 27.25 | 46.52 |
| Ferulic acid | ND * | ND * | ND * |
| Naringenin | 8.25 | 67.52 | 133.42 |
| Taxifolin | ND * | 23.00 | 32.03 |
| Cinnamic acid | 1.16 | 9.21 | 12.47 |
| Kaempferol | 19.92 | 87.08 | 211.36 |

\* Control: without any treatment; BRs: brassinosteroids; CHT: chitosan; ND: not detected.

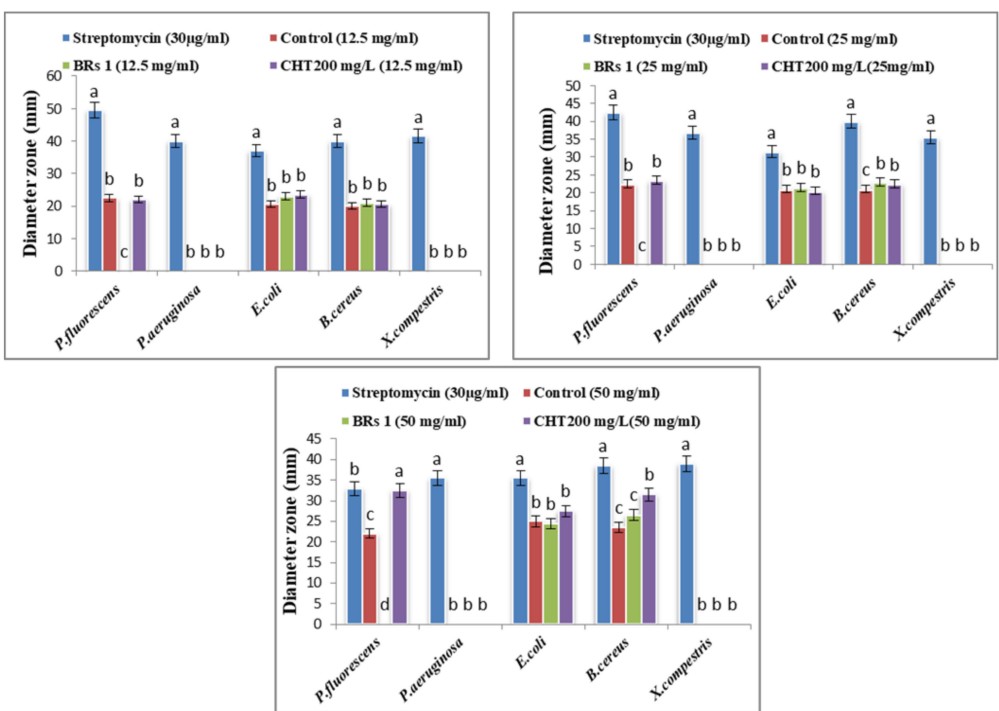

**Figure 1.** Effect of control, BRs (1) at $10^{-8}$ M and CHT 200 mg/L on bacterial activity of *Solidago canadensis* cv. Tara plant. Columns data with different letter/s differ significantly (*p* ≤ 0.05). letters (a–c) are comparison of individual treatment means.

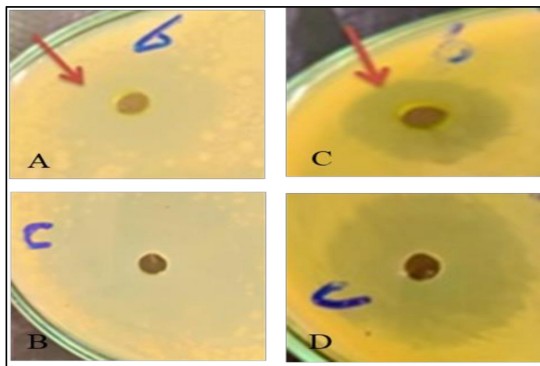

**Figure 2.** Effect of CHT 200 mg/L on different bacterial activity of *Solidago canadensis* cv. Tara plant. (**A,B**): Effect of CHT 200 mg/L on *P. fluorescens*. (**C,D**): Effect of CHT 200 mg/L on *Bacillus cereus*.

In plants, BRs can act efficiently as immunomodulators when used at the correct stage of plant growth and in a suitable concentration. Brassinosteroids are associated in the plant with the response to abiotic environmental stresses, and cause profound alterations in plants that interfere with microbial and viral pathogens. BRs are opening up new approaches for plant resistance against unfriendly environmental conditions [69].

In addition, chitosan has anti-microbial properties and can therefore protect plants against bacterial and fungal pathogens, and benefits the increase in total phenolic and flavonoid contents and compounds in secondary plants. In general, the use of chitosan as a foliar spray and elicitor increased the phenolic compounds, and flavonoids and antioxidant power. This is consistent with chitosan's properties and previous research [57].

*3.6. SRAP-PCR Analysis*

The SRAP (sequence-related amplified polymorphism) technique was developed by Li [15] to identify overlapping coding and non-coding regions of the genome. It is dependent on the amplification of open reading frames (ORFs) using the GC rich exons and the promoter [15]. SRAP not only amplifies the interval between genes and their non-coding flanking regions, but also tightly links to actual genes, which generates a fingerprint of the coding sequences and permits easy isolation of these bands for sequencing [70]. At the molecular level, twelve SRAP primer combinations of four forward and four reverse primers were used in this study. All of the primer combinations successfully tested a total of 115 loci, which were amplified with an average of 9.6 loci/combinations, with molecular sizes ranging from 70 to 1500 bp (Table 7 and Figure 3). The number of genetic loci detected by the SRAP markers ranged from 5 loci in combinations ME1*EM4 and ME2*EM1, to 15 loci in combination ME4*EM2. The percentage of polymorphism ranged from 0% (combinations ME3*EM1 and ME3*EM2) to 88% (combination ME1*EM2). The average level of polymorphism was estimated to be 23%. The polymorphism information content (PIC) values ranged from 0.1 (combinations ME1*EM3 and ME2*EM2) to 1.0 (combinations ME3*EM1 and ME3*EM2), with an average of 0.8. These results demonstrate the presence of genetic variation between the control and treated plants. The SRAP marker was used to study population structure, the genetic linkage map of plants, genetic diversity, genealogical classification, genetic map construction, and cloning [71,72]. The fingerprint genomic cDNA and DNA were from different plants [73]. Our results are in agreement with El-Nashar [74], who also used SRAP markers to study the molecular diversity of *Calendula officinalis* L. after treatment with colchicine as a mutagenic agent. They reported the presence of significant genetic diversity among calendula plants exposed to different concentrations of colchicine. SRAP markers have a high resolution and can discriminate the beneficial effects of colchicine at low concentrations from those of lethal concentrations, which lead to serious genetic errors, and a decrease in growth and the flower parameters. Wu [75] reported that SRAP markers are efficient tools for estimating the genetic relationships and the

genetic variability between buffalo grass populations. Zagorcheva [76] noted that SRAP markers are an effective tool for studying lavender genetic diversity, and demonstrated a wide range of potential for further applications in lavender cultivation and breeding. The dendrogram generated based on the analysis of the SRAP-PCR profile showed that cluster analysis was used to assess the genetic diversity and relationships between the control genotype and the plants treated with brassinosteroids and chitosan, as used in this study. The cluster analysis of two major clusters, i.e., cluster 1 including the control and cluster 2 combining BRs at $10^{-8}$ M and CHT 200m g/L, also revealed low genetic variation among the control and treated plants with brassinosteroids and chitosan (89% and 93%, respectively); the results are shown in Table 8 and Figure 4. Moreover, in plants, brassinosteroids regulate gene expression, nucleic acid, and protein synthesis. Thus, they enhance the elongation and division of the cells. In addition, chitosan can interact with chromatin and directly influence gene expression [33–77].

**Table 7.** Summary of SRAP data following analysis of control, BRs$10^{-8}$ M and CHT 200 mg/L treatments.

| Combination Primer | Total Bands | Polymorphic Bands | Unique Band | Bands of MW * | Polymorphism% | PIC * Value |
|---|---|---|---|---|---|---|
| ME1*EM1 | 7 | 4 | 3 | 70–520 | 57 | 0.7 |
| ME1*EM2 | 8 | 7 | 1 | 80–800 | 88 | 0.7 |
| ME1*EM3 | 7 | 0 | 0 | 70–800 | 0 | 0.1 |
| ME1*EM4 | 5 | 2 | 0 | 70–800 | 40 | 0.9 |
| ME2*EM1 | 5 | 1 | 1 | 80–600 | 10 | 0.9 |
| ME2*EM2 | 7 | 0 | 0 | 90–600 | 0 | 0.1 |
| ME2*EM3 | 6 | 1 | 0 | 100–600 | 17 | 0.9 |
| ME3*EM1 | 10 | 0 | 0 | 160–1000 | 0 | 1.0 |
| ME3*EM2 | 12 | 0 | 0 | 150–1500 | 0 | 1.0 |
| ME3*EM3 | 13 | 1 | 1 | 250–1000 | 8 | 0.9 |
| ME4*EM1 | 10 | 1 | 2 | 200–900 | 30 | 0.9 |
| ME4*EM2 | 15 | 2 | 1 | 150–1600 | 20 | 0.9 |
| Total | 115 | 19 | 9 | - | | |
| Average | 9.6 | 1.6 | - | - | 23 | 0.8 |

* MW, Molecular weight; * PIC, Polymorphism information content; The average of polymorphism = the mean of polymorphism percentage.

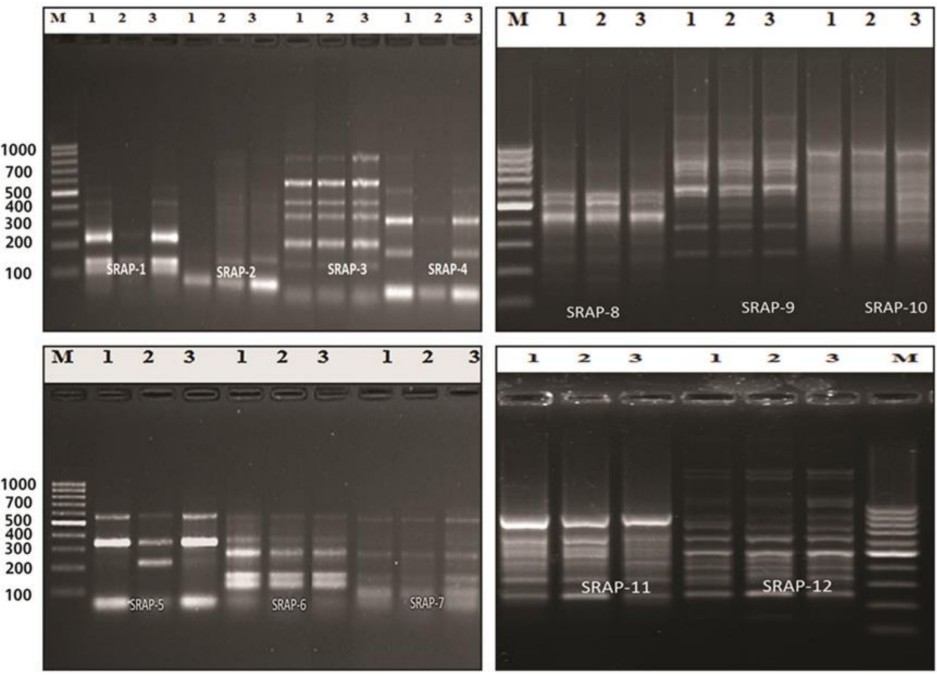

**Figure 3.** PCR amplification using SRAP primer combinations of control, BRs at $10^{-8}$ M, and CHT 200 mg/L (1–3), M (DNA ladder 100 bp) of *Solidago canadensis* cv. Tara plant.

**Table 8.** Similarity matrix based on analysis of SRAP analysis of control and two treatments of *Solidago canadensis* cv. Tara plant.

| | Control | BRs10$^{-8}$ M | CHT 200 mg/L |
|---|---|---|---|
| **Control** | **100** | | |
| BRs at 10$^{-8}$ M | 89 | 100 | |
| CHT 200 mg/L | 93 | 94 | 100 |

Control: without any treatment; BRs: brassinosteroids; CHT: chitosan.

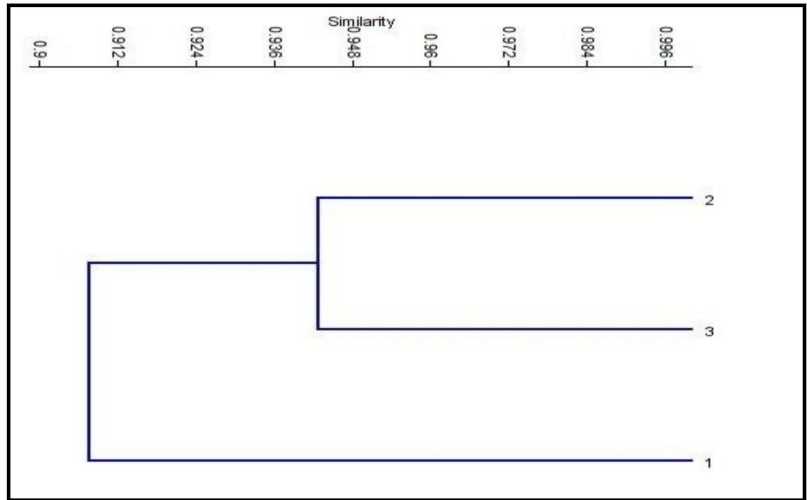

**Figure 4.** Neighbor-joining phylogenetic tree constructed from SRAP data following the analysis of control and two treatments of *Solidago canadensis* cv. Tara plant. 1: Control (without any treatment); 2: BRs at 10$^{-8}$ M; 3: CHT 200 mg/L.

## 4. Conclusions

In brief, these results demonstrate the advantages of applying brassinosteroids and chitosan to *Solidago canadensis* cv. Tara. The optimal ability was found with BRs of 10$^{-8}$ M and chitosan of 200 mg/L for improving growth and flowering parameters. In addition, treatment stimulated the photosynthetic content, N, P, and K content, and total phenolic content. The plant extracts showed high antioxidant capacity and less genetic diversity. This can help enhance the production of goldenrod, thus increasing the country's economic income. The SRAP molecular marker is a powerful and potent technique for studying the genetic diversity in *Solidago canadensis* cv. Tara. Moreover, this study revealed the antibacterial properties of this plant extract, indicating its potential for use as a natural preservative in pharmaceutical, food, and cosmetic products. Further studies will examine the nutritional potential of goldenrod via the implementation of a suitable rat animal model experiment.

**Author Contributions:** Methodology, I.M.E.-S. and D.M.S., Writing—original draft I.M.E.-S. and R.A.E.-Z., Writing—review and editing, I.M.E.-S., R.A.E.-Z., R.G.S., D.M.S., and, E.F.E.-H., All authors have read and agreed to the published version of the manuscript.

**Funding:** This research received no external funding.

**Conflicts of Interest:** The authors declare no conflict of interest.

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
