# Peer review of "Molecular Characterization and Positive Impact of Brassinosteroids and Chitosan on Solidago canadensis cv. Tara Characteristics"

_horticulturae, doi:10.3390/horticulturae6040100_

Round 1

Reviewer 1 Report

Dear Authors,

Natural growth promoters and nutrients are safe and may effect also at the chemical content of medicinal plants. It is may be useful for economical and fast pharmaceutical manufacture as well as in food manufacture and production of polyherbal cosmetic agents.

You did complex interesting investigation about morphological parameters during life cycle, phytochemical analysis, detected molecular polymorphism of samples of Solidago canadensis.

Several comments:

  • during all text, please, check punctuation, capital letters, delete extra spaces and and add where they absence. For example: line 12 - Ornamental; line 35 - "...and Russia) [1]. It is..."; line 39 - "...activities such as antimicrobial,...[4-5]." (delete ; and brackets for references).
  • line 40 - Did you mean flavonoids and phenolic acids? Here's solidago must be at capital letter and Italic style;
  • lines 61-63 - abbreviations should be after the decryption;
  • Table 1 - "EC ds/m" and "Meq/l" - please, give description and write "l" as "L";
  • chapters 2.6-2.9 -in what plant part you measures amount of pigments, K, P and N?
  •  chapter 2.11 - please, described dose of streptomycin as standard;
  • lines 176 and 419 - different decryption of genetic similarity coefficient;
  • line 187 - mg/L, not mg/l;
  • Tables 3 and 4 - number of leaves/shoots (table 3) and No of inflorescence/plant (table 4) - it is not integral number?
  • line 265 - Xanthium and Rumex - italic style;
  • line 350 - caffeic acid, not coffeic. In Table please, also correct;
  • line 353 - rutin is not polyphenolic component. Only catechin, methyl gallate, pyrocatechol, ellagic acid can be attributed at polyphenols;
  • Table 6 - in line "Polyphenolic compounds content of Solidago canadensis...", please, correct capital letter in species name;
  • line 383 - EO - what did you mean?
  • Fig. 1 - please, describe units on the ordinate axis;
  • Table 7 - please, describe decryption of MW and PIC and line Average in column polymorphism % - delete % in "%23";
  • lines 478-479 - why you wrote about essential oil? During the manuscript there were  not data about essential oil from S. canadensis

Author Response

Response to Reviewer 1 Comments

Point 1: during all text, please, check punctuation, capital letters, delete extra spaces and add where they absence. For example: line 12 - Ornamental; line 35 - "...and Russia) [1]. It is..."; line 39 - "...activities such as antimicrobial,...[4-5]." (Delete; and brackets for references).

Response 1: we checked all punctuation, capital letters, delete extra spaces and add where they absent.

Point 2: line 40 - Did you mean flavonoids and phenolic acids? Here's solidago must be at capital letter and Italic style;

Response 2: yes we meant flavonoids and phenolic acids and I wrote solidago at capital letter and Italic style.

Point 3: lines 61-63 - abbreviations should be after the decryption;

Response 3: we wrote the abbreviations after the decryption.

Point 4: Table 1 - "EC ds/m" and "Meq/l" - please, give description and write "l" as "L";

Response 4: we wrote the description of "EC ds/m" and "Meq/l"(Electrical conductivity (EC)- deciSiemens per metre (dS/m)- milliequivalents per litre (Meq/L) and wrote "l" as "L";

Point 5: chapters 2.6-2.9 -in what plant part you measures amount of pigments, K, P and N?

Response 5: the plant part of pigments leaves but NPK is dried herb.

Point 6: chapter 2.11 - please, described dose of streptomycin as standard;

Response 6: we described the dose at manuscript.

Point 7: lines 176 and 419 - different decryption of genetic similarity coefficient

Response 7: Line 176 decryption the method of SRAP analysis 419 is the decryption of SRAP as molecular marker not analysis.

Point 8: line 187 - mg/L, not mg/l;

Response 8: we changed of all them at manuscript

Point 9: Tables 3 and 4 - number of leaves/shoots (table 3) and No of inflorescence/plant (table 4) - it is not integral number?

Response 9: yes number of leaves/shoots and No of inflorescence/plant are integral number but the numbers in tables are an average. We change it to integral number.

Point 10: line 265 - Xanthium and Rumex - italic style;

Response 10: we confirm it to italic.

Point 11: line 350 - caffeic acid, not coffeic. In Table please, also correct;

Response 11: we confirm caffeic to coffeic.

Point 12: line 353 - rutin is not polyphenolic component. Only catechin, methyl gallate, pyrocatechol, ellagic acid can be attributed at polyphenols;

Response 12: we corrected it.

Point 13: Table 6 - in line "Polyphenolic compounds content of Solidago canadensis...", please, correct capital letter in species name;

Response 13: we corrected capital letter in species name.

Point 14: line 383 - EO - what did you mean?

Response 14: EO means essential oil.

Point 15: Fig. 1 - please, describe units on the ordinate axis;

Response 15: we described units on the ordinate axis.

Point 16: Table 7 - please, describe decryption of MW and PIC and line Average in column polymorphism % - delete % in "%23";

Response 16: we described decryption of MW and PIC and line Average in column polymorphism % and deleted % in "%23.

Point 17: lines 478-479 - why you wrote about essential oil? During the manuscript there were not data about essential oil from S. canadensis.

Response 17: we removed it.

Reviewer 2 Report

Dear colleagues, it would be interesting to evaluate in your future works whether there could be a synergistic effect of brassinosteroids and chitosan as well as on morphological characteristics, in particular on the evaluation of antioxidant capacity. Another useful aspect to consider for the production of supplements is to investigate whether different concentrations of chitosan can counteract the phenomenon of antibiotic resistance.

Author Response

Response to Reviewer 2 Comments

Point 1: Dear colleagues, it would be interesting to evaluate in your future works whether there could be a synergistic effect of brassinosteroids and chitosan as well as on morphological characteristics, in particular on the evaluation of antioxidant capacity. Another useful aspect to consider for the production of supplements is to investigate whether different concentrations of chitosan can counteract the phenomenon of antibiotic resistance.

Response 1: Thanks for your concern and response to our paper. In the future, we will evaluate a synergistic effect of brassinosteroids and chitosan as well as on morphological characteristics, in particular on the evaluation of antioxidant capacity.

Reviewer 3 Report

The manuscript by Iman M. El-Sayed  and co-autors titled “Molecular Characterization and Positive Impact of Brassinosteroids and Chitosan on Solidago canadensis cv. Tara Characters” is interesting, the research results are practical and can be used in horticultural production. However, the publication requires the introduction of necessary corrections and supplementations indicated below for the Author’s consideration. Specific suggestions and requests are included in the text, inserting them after highlighting the wordings to which they refer. In my opinion the paper is acceptable for publication in Horticulturae  after revision.

Author Response

Response to Reviewer 3 Comments

Point 1: These are already present in the title.

Response 1: Brassinosteroids and chitosan are already present in the title so, we deleted it from key words.

Point 2: please specify the temperature range

Response 2: we specified the temperature range.

Point 3: by which method the leaf area was tested?

Response 3: we added the method in the manuscript and ref.

Point 4: Please describe in detail the methods for the determination of chlorophyll and carotenoids. Literature number 17 is not available to everyone.

Response 4: The method of chlorophyll and carotenoids was written in the paper.

Point 5: You present some of your results as a two-year average and some of your results as averages for each season. What is your explanation of this? It would be good to present all the results for each season separately.

Response 5: we mention morphological parameter for each season but for chemical parameter we take a sample as an average for each season because the chemical which uses to the determination that (cost a lot of money, so we take it as average).

Point 6: Why the identification of phenolic compounds was done only for the control and for the highest concentrations of BRs and CHT?

Response 6: Because the selected treatments of BRs and CHT are given the best results of all parameters. So, I determined in these treatments compared with control.

Point 7: Is it statistically confirmed? Some differences between means were not statistically significant, please correct. See table.

Response 7: yes we saw it and correct it in results.

Point 8: Please discuss the results of the experiment with world references in more detail.

Response 8: we discussed it in more detail at manuscript.

Point 9: This is not related to the aspect of yours experiment.

Response 9: we removed it manuscript.

Point 10: Please discuss the results of the experiment with world references in more detail.

Response 10: we discussed the results of the experiment with world references in more detail at manuscript.

Point 11: Some differences between means were not statistically significant, please correct. See table.

Response 11: we corrected it.

Point 12: But not compared to the control

Response 12: we compared it to the control.

Point 13: Maybe "content of macronutrients"

Response 13: we changed it to content of macronutrients

Point 14: Only for N. See means in Table 5

Response 14: yes we corrected it to subscribe like another letters.

Point 15: What is your explanation of this?

Response 15: we explanted it at manuscript

Point 16: not determined in all combination - can be removed

Response 16: it is not detected but not determined.

Point 17: What is your explanation of this?

Response 17: we explanted it at manuscript.

Point 18: Really, is it such important species for the macroeconomy?

Response 18: yes it is crucial species for the macroeconomy because with increasing the production the macroeconomy will increase (export)

Point 19: This is true, but was they that important in this research?

Response 19: using SRAP molecular marker showing the variation between best treatments compared to control.
